# Microglia: The Drunken Gardeners of Early Adversity

**DOI:** 10.3390/biom14080964

**Published:** 2024-08-08

**Authors:** Sahabuddin Ahmed, Baruh Polis, Arie Kaffman

**Affiliations:** Department of Psychiatry, Yale University School of Medicine, 300 George Street, Suite 901, New Haven, CT 06511, USA; sahabuddin.ahmed@yale.edu (S.A.); baruhpolis@gmail.com (B.P.)

**Keywords:** early adversity, microglia, maternal separation, limited bedding and nesting, synaptic pruning, neurodevelopment

## Abstract

Early life adversity (ELA) is a heterogeneous group of negative childhood experiences that can lead to abnormal brain development and more severe psychiatric, neurological, and medical conditions in adulthood. According to the immune hypothesis, ELA leads to an abnormal immune response characterized by high levels of inflammatory cytokines. This abnormal immune response contributes to more severe negative health outcomes and a refractory response to treatment in individuals with a history of ELA. Here, we examine this hypothesis in the context of recent rodent studies that focus on the impact of ELA on microglia, the resident immune cells in the brain. We review recent progress in our ability to mechanistically link molecular alterations in microglial function during a critical period of development with changes in synaptic connectivity, cognition, and stress reactivity later in life. We also examine recent research showing that ELA induces long-term alterations in microglial inflammatory response to “secondary hits” such as traumatic brain injury, substance use, and exposure to additional stress in adulthood. We conclude with a discussion on future directions and unresolved questions regarding the signals that modify microglial function and the clinical significance of rodent studies for humans.

## 1. Introduction

Early life adversity (ELA) encompasses various childhood hardships such as neglect, abuse, severe poverty, or exposure to high levels of neighborhood crime [1,2,3,4]. ELA is recognized as one of the most important and preventable causes of abnormal brain development and psychopathology later in life [4,5]. The Child Maltreatment Report issued by the Department of Health and Human Services documented 600,000 cases of ELA in 2021 alone [6]. Most of these cases were due to neglect (76%), followed by physical abuse (16%), and sexual abuse (10%). Two-thirds of the population in the US experienced at least one form of adversity, and one-sixth reported exposure to at least four different types of adversities [4]. This is important because exposure to a greater number of adversities increases the risk of developing multiple medical and psychiatric conditions later in life [7,8,9]. This observation has led some to propose the “cumulative model”, in which the number of adversities is simply added up to quantify ELA severity and assess the risk for psychopathology [10]. A couple of years later, proponents of the “dimension model” challenged the assertion that all adversities cause similar outcomes that can be added up [11,12]. Instead, they proposed that adversities fall into two distinct dimensions that lead to different developmental outcomes. One dimension involves adversities such as physical or sexual abuse, which share a common exposure to different levels of physical threat (threat dimension). A second dimension, particularly childhood neglect and extreme poverty, is characterized by the absence of appropriate sensory, cognitive, and social stimulation (deprivation dimension). They note that adversities within the threat dimension are commonly associated with mood dysregulation and abnormal threat detection, whereas deprivation is more commonly associated with cortical thinning, hyperactivity, and autistic-like presentation [11,12,13]. Individuals are often exposed to multiple adversities across dimensions, making it challenging to test the validity of the cumulative and dimensional models, leading to inconsistent results in the field [14].

Rodent models of ELA can control for the timing, type, severity, and genetic background. Therefore, they can provide crucial insights into how deprivation and threat impact neurodevelopment individually and when combined in the same individual [14,15]. The use of imaging tools such as diffusion MRI (dMRI) and resting-state fMRI (rsfMRI) in rodents provides an important translational strategy to examine outcomes in an unbiased and non-invasive manner. These outcomes can then be compared with those found in humans [15,16]. Such an approach can also be used to develop improved diagnostic tools and evaluate treatment responses. Rodent models can also elucidate the molecular and cellular changes induced by different adversities and establish causal links with behavioral and cognitive deficits [15,17,18].

In other words, animal models may elucidate important details about the underlying developmental changes that are responsible for the devastating impact of ELA on brain development, leading to better diagnostic tools and interventions. These interventions are desperately needed because approximately half of all childhood psychiatric diagnoses are attributed to ELA [19], and in many cases, these persist as chronic and refractory psychopathology later in life [5]. For example, 44–54% of all cases of adult major depression in the US are attributed to ELA [20,21] and a meta-analysis by Nani et al. (2012) found that depression in individuals with a history of ELA is characterized by more frequent depressive episodes, increased duration, and reduced responsivity to treatment [22]. A similar meta-analysis also identified a more severe and refractory subtype of bipolar disorder in individuals with a history of ELA [23]. ELA is also associated with increased levels of inflammatory markers such as CRP, IL-6, and Tnfα [24,25], leading multiple researchers to propose that an abnormal immune response plays a crucial role in mediating the more severe and unique features of psychopathologies associated with ELA [5,26,27]. Studying this hypothesis in humans is challenging due to the complex nature of the adversities, genetic variability, and our inability to access and manipulate the human brain. However, it represents an important new frontier in animal research. Although excellent reviews on this topic are available [28,29,30], the rapid progress in recent years has compelled us to outline key advancements focusing on reproducible findings that causally link ELA-induced changes in microglia, the brain’s specialized macrophages, with alterations in neurodevelopment and psychiatrically relevant changes later in life. 

The review is presented in three sections. The first section outlines the rationale for focusing on microglia as potential cellular substrates for altering brain development in response to various types of ELA. The second section reviews recent research implicating microglia in this process, focusing on deficits in synaptic pruning and heightened inflammatory responses to secondary hits in adulthood. The third section discusses key unanswered questions and future directions. 

## 2. The Rationale for Focusing on Microglia

Microglia are phagocytic immune cells that enter the rodent brain at embryonic day 9.5 (E9.5) where they regulate multiple developmental processes, such as neurogenesis, neuronal migration, removal of apoptotic or damaged cells, axonal growth, synaptogenesis, synaptic pruning, vascular development, and myelination [28,31,32,33]. During early development, from E9.5 to postnatal day 14 (P14), microglia divide and migrate as amoeboid cells to evenly distribute themselves across the entire brain parenchyma. Upon arrival at their destination, they extend dynamic “cellular tentacles” equipped with numerous receptors that allow them to monitor a large number of signals and rapidly guide development [32,33], Figure 1A. Microglia alter development by secreting cytokines and neurotrophic factors that promote the division, survival, and differentiation of neurons and other cells in the brain. In addition, through dynamic cellular processes, they can recognize and remove excessive or non-functional cellular debris that interferes with appropriate brain development [32,33]. One of the best examples of this is synaptic pruning, during which receptors such as TREM2 and MerTK expressed on microglial processes recognize “eat-me signals” on non-functional synapses and remove them during a specific developmental period [34,35,36,37,38]. Synaptic pruning ensures the formation of a more mature synaptic grid and efficient connectivity later in life (Figure 1A). Conditions that impair microglial-mediated synaptic pruning during development result in long-term deficits in synaptic maturation, connectivity, and behavior [34,38,39,40,41,42,43]. Similarly, sensory deprivation early in life, such as that caused by whisker trimming [44] or monocular deprivation [45], leads to long-term cortical changes and sensory deficits that are difficult to reverse in adulthood [28,46].

Receptors expressed on microglial “cellular tentacles” survey the surrounding brain tissue at an unparalleled speed [47] making these cells highly sensitive to detecting environmental changes associated with ELA. These include mediators of stress/threat, sensory, and cognitive deprivation commonly seen in neglect, and changes in nutritional status and the gut microbiome [28,48,49]. Microglia’s exquisite sensitivity in detecting environmental changes, coupled with their ability to influence various developmental processes, make them compelling cellular candidates for facilitating developmental changes associated with ELA [28]. Indeed, the recent work described in Section 3.1 and Section 3.2 below rigorously demonstrates how alterations in microglial-mediated synaptic pruning during a critical period of development leads to long-term changes in connectivity and behavior. Finally, some forms of ELA alter microglial responses to “secondary hits” later in life, such as traumatic brain injury [50], substance use [51], or additional stress [52]. This heightened inflammatory response worsens psychiatric and neurological outcomes later in life and may require different treatment for those exposed to ELA. The next section highlights significant progress in establishing a causal link between ELA, abnormal microglial function, and developmental and behavioral outcomes later in life.

## 3. Rodent Studies 

ELA alters microglial number [50,51,53,54], morphology [36,39,50,53,55,56,57,58], surveying dynamics [38,59], phagocytic activity [36,38,39,55,58], and gene expression [36,53,58] during development. However, relatively few findings have been replicated, and most studies to date have not causally linked changes in microglial function with alterations in neurodevelopment, connectivity, or behavior. In this section, we focus on research that causally links changes in microglia with alterations in neurodevelopment (Section 3.1 and Section 3.2) and broad conceptual themes supported by studies from multiple groups. Section 3.3 reviews data indicating that early threat and deprivation affect microglial function differently, while Section 3.4 discusses how perturbation of microglial function leads to different outcomes during development and adulthood. Section 3.5 summarizes studies that elegantly demonstrate how ELA alters microglial response to secondary hits and the impact of this abnormal response on functional outcomes in adulthood.

### 3.1. Limited Bedding Transiently Impairs Microglial-Mediated Synaptic Pruning during a Critical Period of Hippocampal Development Leading to Sex-Specific Deficits in Adolescent Mice

Abnormal hippocampal development and function are consistently observed in individuals exposed to ELA, with some evidence indicating that men may be more severely affected than women [1,14,30]. Similar observations in rodent models of ELA suggest that research in rodents may provide crucial new insights into the underlying mechanisms [39,60]. We have recently shown that adolescent male mice exposed to extended limited bedding (LB) from P0–25 exhibited severe deficits in contextual fear conditioning, which is a commonly used test to assess hippocampal function. Female LB littermates showed only a minor impairment that was not statistically significant compared to control females [39,60]. These sex-specific cognitive deficits were associated with the retention of a large number of immature spines and a reduction in the number of mature (mushroom) spines in the apical dendrites of CA1 neurons located in the stratum radiatum, which were more pronounced in LB males [39]. We defined the synaptic maturity index as the ratio between mature and immature spines and demonstrated a significant correlation with contextual fear conditioning [39]. Using high-resolution confocal microscopy, we further confirmed a reduced density of glutamatergic synapses in the stratum radiatum in LB males, but not in LB females [39]. These findings are consistent with research indicating that synaptic connections in the stratum radiatum play a critical role in hippocampal-dependent memory [61,62,63,64,65]. Additionally, using resting-state functional MRI (rsfMRI), we observed that LB reduced local functional connectivity across several brain regions, including the hippocampus, entorhinal cortex, hypothalamus, and the amygdala in adolescent males, but not in LB females [39]. The findings align with the more pronounced synaptic and hippocampal-dependent memory deficits observed in LB male mice and indicate that sex-specific effects on connectivity extend beyond the hippocampus.

Given that microglial-mediated synaptic pruning peaks in the developing hippocampus during the second and third weeks of life [34,35,39,41,42,66], a process that ensures effective connectivity and normal hippocampal-mediated function later in life [39,41], we hypothesized that LB impairs microglial-mediated synaptic pruning in the developing hippocampus of male, but not female, pups. This was not the case, and we found that LB impaired microglial-mediated synaptic pruning in both male and female P17 pups [36,39]. LB did not affect the number of microglia but severely reduced microglial cell volume, ramification, phagosome size, and the ability to engulf synaptic material [36,39], Figure 1B–D. We replicated our findings using an ex vivo system, demonstrating that freshly isolated microglia from the hippocampus of P17 LB mice had a reduced capacity to phagocytose pHrodo-labeled synaptosomes [36]. Additionally, we found that LB decreased the expression of the TREM2 receptor on microglia and that a reduction in this receptor led to similar morphological and phagocytic abnormalities observed in LB mice [36]. This is important because previous work has shown that the microglial TREM2 receptor is essential for normal synaptic pruning in the developing hippocampus, as well as for normal social behavior and connectivity later in life [34,35]. Finally, deficits in microglial-mediated synaptic pruning were detected in the hippocampus of P17 pups, but not in adolescent P33 mice, when the levels of synaptic pruning are approximately 8-fold lower [36,39], Figure 1B–D.

Next, we developed a system that allows us to transiently ablate microglia during the second and third weeks of life. We found that transient ablation of microglia causes similar sex-specific abnormalities as those observed in LB mice. These abnormalities include deficits in contextual fear conditioning, reduced synaptic maturity index, decreased glutamatergic synapse density, and lower local functional connectivity measured using rsfMRI [39]. Furthermore, chemogenetic activation of microglia at P13-17 was able to rescue the phagocytic deficits detected in P17 LB microglia and normalize the cognitive and synaptic abnormalities observed in LB males [39], demonstrating that normalizing microglial function during a crucial developmental window alleviates some of the negative impacts of ELA on brain connectivity and cognition. 

Despite experiencing similar abnormalities in microglial phagocytic activity, adolescent LB females exhibited only minimal deficits in hippocampal-dependent memory and synaptic strength [39,60]. This sex-specific phenomenon prompted us to examine the role of astrocytes, another vital glial cell involved in synaptic pruning [67,68,69,70,71]. Indeed, we found that 17-day-old LB females, but not LB male littermates, displayed increased astrocyte-mediated synaptic pruning. The differences in astrocytic function were associated with the upregulation of the MEGF10 receptor in the astrocytes of LB females, an increase that was not seen in astrocytes from LB males [39]. Since the MEGF10 receptor is necessary for normal synaptic pruning in astrocytes [67,71], we propose that LB females can upregulate MEGF10 receptors on astrocytes. This, in turn, allows LB females to enhance synaptic pruning in astrocytes and minimize deficits in microglial-mediated synaptic pruning.

Overall, the above studies demonstrate that LB impairs microglial-mediated synaptic pruning during peak synaptic pruning in the developing hippocampus by reducing TREM2 expression on microglia (Figure 1C). The removal of excessive non-functional synapses during this period is thought to redirect limited neuronal resources such as Ca^2+^/calmodulin-dependent protein kinase II [72] to functional synapses leading to an increase in the number of mushroom spines compared to immature spines (Figure 1A). This increase in the synaptic maturity index is necessary to enhance local functional connectivity and hippocampal function during the adolescent period (Figure 1C). Transient perturbation of microglial function during this critical period is responsible for abnormal hippocampal function in adolescent LB males, while upregulation of synaptic pruning in astrocytes may protect female mice [39].

Although our work has focused on microglial dysfunction in the developing hippocampus, a significant reduction in microglial size is observed across many brain regions in P17 LB mice. Further, abnormal local functional connectivity is observed across many brain regions in male mice exposed to LB or after transient ablation of microglia [39], suggesting that the deficits in microglial-mediated synaptic pruning are not specific to the developing hippocampus.

### 3.2. Microglial-Mediated Synaptic Pruning of Corticotropin-Releasing Hormone (CRH) Neurons in the Developing Hypothalamus Programs Stress Reactivity Later in Life

Levels of maternal care during the first week of life bidirectionally program stress reactivity in adult rodents [38,73,74,75]. For example, exposure to limited bedding and nesting (LBN) from P2-9 causes erratic maternal care during the first week of life and leads to increased stress reactivity and abnormal threat detection in adulthood [38,73]. In contrast, rodents exposed to brief handling, which leads to augmented maternal care, display a blunted response to stress and reduced anxiety-like behavior later in life [74,75]. This bidirectional relationship seems to be influenced by the quantity of glutamatergic synaptic inputs innervating CRH-positive neurons in the paraventricular nucleus of the hypothalamus. Higher levels of maternal care are associated with reduced glutamatergic innervation, which, in turn, increases the expression of the neuron-restrictive silencer factor (NRSF). NRSF binding to multiple promoters, including the CRH promoter, induces a stable repressive chromatin structure that inhibits CRH expression throughout life [74,75]. Since the secretion of CRH from these neurons is necessary to initiate the hypothalamic–pituitary–adrenal (HPA) response to stress [76], the reduction in CRH expression is responsible for the blunted stress reactivity observed in offspring exposed to higher maternal care [74,75]. Recent work by Bolton and colleagues (2022) has shown that impaired microglial-mediated synaptic pruning during the first week of life is responsible for the increased glutamatergic innervation of CRH-positive cells observed in mice exposed to LBN [38]. More specifically, they have shown that microglial-mediated synaptic pruning of CRH-positive neurons in the PVN peaks at P8 and is reduced by five-fold in P24 juvenile mice. LBN decreases microglial process motility and reduces levels of the MerTK receptor on microglia. Impaired synaptic engulfment during this critical period leads to the retention of many glutamatergic inputs on CRH-positive neurons. This increase in synaptic input persists into adulthood and is linked to heightened stress reactivity and abnormal threat detection in the looming object test [38]. Chemogenetic activation of microglia increased process motility in microglia from LBN but had no impact on microglial motility in control mice. Moreover, chemogenetic activation of microglia from P3-10 was able to normalize glutamatergic innervation of CRH-positive cells at P10 and restore normal stress reactivity and threat detection in adult LBN mice [38]. Two additional important findings from this work include the observation that only microglia in the vicinity of CRH-positive cells were impacted, indicating that some molecular signals, perhaps CRH, secreted by these cells inhibit the ability of microglia to remove glutamatergic inputs from these cells [38]. Secondly, microglial-mediated synaptic pruning of CRH-positive cells was impaired in male pups, but not in female littermates, further highlighting significant sex differences in microglial response to ELA [30,38].

### 3.3. Different Types of Adversities Cause Different Changes in Microglial-Function

The prolonged impoverished conditions that characterize the LB paradigm are likely to induce high levels of deprivation with relatively low levels of threat [14,36]. In support of this prediction, we recently showed that LB causes significant cortical thinning and hyperactivity, outcomes that are typically seen in children with severe deprivation and neglect [60]. Maternal separation, another commonly used rodent model of ELA, exposes pups to higher levels of threat and HPA activation because pups are fully dependent on the mother for survival during the first two weeks of life [14,36]. The maternal separation paradigm also differs from the LB paradigm because it leads to a compensatory increase in maternal care and exposes pups to a novel environment [14,36]. Therefore, even though LB and maternal separation are both models of ELA, they represent different types of adversities that may not impact microglia or neurodevelopment in the same way. Indeed, exposure to limited bedding leads to a significant reduction in microglial ramification and cell volume in the developing hippocampus [36,39,56,57] whereas maternal separation increases microglial ramification and volume during the same period [36,53,55]. This is not a surprising result but one that is frequently ignored when discussing inconsistent outcomes or when considering ELA as a single entity rather than a heterogeneous group of hardships.

As discussed above, many individuals are exposed to multiple adversities, and it is currently unclear whether exposure to multiple adversities can be summed up as proposed by the cumulative model or considered separately based on the dimensional model. We investigated this question by comparing outcomes in mice exposed to no adversities (e.g., control condition), mice exposed to a single adversity (e.g., limited bedding, or LB), and mice exposed to two stressors (LB and unpredictable maternal separation, or UPS). UPS mice were raised with limited bedding but were also separated from the dam for 1 h on P14, 16, 17, 21, 23, and 25 [36]. We first investigated the impact of these three rearing conditions on corticosterone blood levels at P17 and found higher corticosterone levels in UPS compared to control mice, with LB showing an intermediate phenotype. This finding is consistent with the cumulative model and the expected increase in threat associated with maternal separation [14]. However, when assessing microglial cell size and phagocytic activity, we found that LB caused more severe deficits compared to UPS microglia which exhibited an intermediate phenotype between the LB and control conditions [36]. Microglia isolated from the hippocampus of UPS mice exhibited a distinct upregulation of several genes implicated in phagocytic activity compared to LB and control groups. Some examples include upregulation of complement genes, CX3CR1, MerTK, Sirpα, and TLR7. Microglia isolated from LB mice had significantly higher mRNA levels of Tnfα and reduced expression of NGF compared to control condition, changes that were reversed by UPS [36]. Adolescent UPS male mice had less severe deficits in contextual fear conditioning compared to adolescent LB male mice (Appendix A). In summary, adding a second adversity in the form of maternal separation shifted the microglial function in LB mice towards a control-like phenotype and reduced cognitive and synaptic deficits later in life. These findings support the dimensional model arguments that different adversities cause different developmental outcomes that are not necessarily additive [36].

### 3.4. Perturbation of Microglial Function Leads to Different Outcomes in Development and Adulthood

Perturbation of microglial function during development and adulthood leads to different outcomes [28]. For example, transient ablation of microglia in the prefrontal cortex of P42 adolescent mice causes severe deficits in working memory, reduced glutamatergic synapse density and a decrease in the density of mature mushroom spines in adulthood [43]. These cognitive and synaptic changes were not observed when microglia were ablated in adulthood, indicating that perturbation during a specific period of development is necessary [43]. Furthermore, exposure to severe stress in adulthood, commonly observed in mice exposed to repeated social defeat, induces a surge of pro-inflammatory cytokines in microglia that contributes to anxiety-like behavior [77,78]. Significant pro-inflammatory cytokine production was also observed in adult microglia in response to TBI [79] and neurodegenerative diseases [80]. In contrast, prominent microglial-mediated inflammation is not commonly seen during development in mice exposed to different types of ELAs [29,36,53,58], with one notable exception [81] that has not yet been replicated. These differences are due to the unique role that microglia play in guiding synaptic pruning during a critical period of development [34,35,36,37,38]. Other factors that may also contribute to the delayed inflammatory response in adulthood include the relative immaturity of the HPA during the first two weeks of life [76,82], greater neuronal injury, and maturation of peripheral and microglial immune responses later in life [77,80,83]. The critical point is that perturbing microglia at different ages leads to different outcomes.

### 3.5. Changes in Microglial Function Persist into Adulthood and Contribute to Long-Term Negative Sequela of Secondary Hits

A substantial body of research has shown that some forms of ELA alter microglial response to “secondary hits” in adulthood. This immune response is not seen in individuals with no history of ELA and exacerbates psychiatric and neurological outcomes associated with such secondary hits [50,51,52,58]. Such findings may explain why individuals with ELA frequently present with more severe psychopathology and exhibit different therapeutic responses compared to individuals with no history of ELA [5]. It also raises the relatively unexplored possibility that ELA may alter the trajectory and outcomes of neurodegenerative diseases such as Alzheimer’s and Parkinsonism [57], where microglia play a prominent causative role [84,85]. Finally, it highlights the necessity of developing specific diagnostic and therapeutic interventions to address this unique form of psychopathology.

For example, Catale et al. (2021) showed that a 30 min daily exposure to a gonadectomized adult CD1 male from P14-21 exacerbates neurological deficits induced by a traumatic brain injury (TBI) in adulthood [50]. Exposure to social stress early in life by itself had no impact on neurological function but required a secondary hit in the form of TBI. The exacerbated neurological outcomes in individuals with ELA were due to more extensive neuronal cell death and an increased microglial inflammatory response mediated by elevated expression of the inflammasome pathway, such as NLRP3 and IL-1β. Administration of the CSFR1 antagonist (GW2580) during the P14-21 social stress blocked microglial inflammation in response to TBI in adulthood. It also reduced levels of neuronal cell death and neurological deficits to the levels observed in non-ELA mice exposed to the same TBI [50]. Administration of GW2580 soon after the brain injury had no impact on microglial inflammation, neuronal cell death, or neurological sequelae, suggesting that interventions early in life might be necessary [50]. Additional studies are needed to clarify how this form of ELA affects microglial responses to TBI later in life and how GW2580, an atypical CSFR1 antagonist that does not deplete microglia [86], restores normal microglial responses to TBI in individuals with a history of ELA.

Work from the same group indicated that this paradigm also exacerbates cocaine-induced contextual place preference in adulthood [51]. Treatment with minocycline from P14-21 reversed the increase in microglial cell number and soma size observed in the ventral tegmental area in P22 mice exposed to ELA and eliminated cocaine-induced conditioned place preference in adult mice with a history of ELA [51]. These findings support the notion that long-term changes in microglial function contribute to increased vulnerability to substance abuse in individuals with a history of ELA. However, the non-specific impact of minocycline makes it difficult to determine whether the observed outcomes were due to changes in microglial response to cocaine use in adulthood.

Han et al. [52] demonstrated that 3 h daily maternal separation during the first two weeks of life did not lead to an increase in anxiety and depression-like behaviors in young adult male mice (P42-56). However, the introduction of a secondary stressor in the form of daily 2 h restraints for 14 days resulted in a robust increase in anxiety and depression-like behaviors in mice with a history of ELA, but not in mice without ELA [52]. Exposure to a secondary hit also reduced microglial ramification in the dentate gyrus and increased levels of several inflammatory cytokines in the hippocampus (e.g., IL-6, IL-1β) in mice with a history of ELA. Increased inflammation was associated with reduced neurogenesis and BDNF levels in the hippocampus. Importantly, the administration of minocycline before the secondary hit (P28-42) prevented the increase in anxiety and depression-like behaviors, the rise in inflammation, and the reduction in neurogenesis and BDNF in mice exposed to ELA and a subsequent stressor [52]. These findings are intriguing, considering clinical data suggesting that minocycline augmentation may be an effective strategy for some forms of treatment-resistant depression [87,88] and that ELA increases the risk of treatment-resistant depression [5,22].

Reemst et al. [58] took a significant step forward in characterizing the long-term impact of exposure to LBN from P2-9 on microglia in the hippocampus of adult males (P200). LBN affected microglial morphology, reduced their capacity to phagocytose synaptosomes ex vivo, and altered gene expression in 186 genes, most notably increasing TNFα-responsive genes and genes implicated in cytoskeleton dynamics [58]. Administration of LPS led to distinct changes in microglia isolated from the adult hippocampus of mice exposed to LBN and control conditions [58]. These findings further support the notion that ELA alters microglial responses to a secondary immunological challenge in adulthood.

## 4. Future Directions

Although significant progress has been made in identifying ELA-induced aberrations in microglial function during development and in adulthood and linking them with physiological, cognitive, and behavioral abnormalities later in life, several key unanswered questions and technical challenges remain (Figure 2). For example, even with some of the best-characterized examples described above, the signals that drive changes in microglial morphology and function have not been identified yet (Figure 2A). In the case of the localized changes observed in the vicinity of CRH-positive neurons at P8 (Section 3.2), such a signal might be CRH itself. However, in the case of the extended LB paradigm described in Section 3.1, the microglial abnormalities, especially the reduced arborization, appear across the entire brain, suggesting a global signal. Abnormal levels of metabolic mediators, such as short-chain fatty acids or other signals from the gut microbiome, might play a role [49,89,90]. Alternatively, the impoverished and deprived conditions that characterize the LB paradigm may lead to abnormal neuronal activation and inadequate neuronal signals, such as ATP, CX3CL1, or IL-33, essential for normal microglial ramification and synaptic pruning. Addressing this question will likely provide additional insights into the mechanisms that drive the expansion of microglial cell size and ramification during the second week of life in normally developing mice (Figure 1A). Multi-omics unbiased approaches to characterize microglia, such as microglial-specific ribotag RNA-seq, single-cell RNA-seq, ATAC-RNA-seq, small-scale proteomics, and spatial transcriptomics, have rarely been used to date. These approaches are likely to identify environmental signals that alter microglial function and reveal stable epigenetic changes making these cells more reactive to secondary hits later in life (Figure 2B,C).

Although the focus of this review has been on the impact of ELA on microglial-mediated synaptic pruning during the postnatal period, additional studies are needed to examine the impact of ELA during the preweaning period on microglial ability to guide vascular development and neural stem cell proliferation, migration, survival, and differentiation [32,33]. Further research is also required to understand the role that microglia play in mediating the deleterious effects of ELA on myelination, blood–brain barrier integrity, and adult neurogenesis (Figure 2C).

Linking molecular changes in microglia with the observed consequences of ELA is hindered by the lack of tools for manipulating gene expression in microglia. Unlike other cells in the brain, microglia are particularly challenging to efficiently and rapidly transfect in vivo using viral constructs [91,92]. Although some progress has been made in addressing this challenge [92,93,94,95,96], no group has yet used this approach to alter gene expression in the developing brain. This approach will also enable researchers to investigate how manipulating microglial gene expression in a specific brain region affects circuit development and function later in life in normally developing mice and in mice exposed to ELA.

Perhaps the most important and challenging task ahead is to test the relevance of findings obtained in rodent models of ELA with outcomes observed in humans. This task is challenging due to the complexity and heterogeneity of ELA in humans, confounding variables such as substance use and medication, the absence of imaging tools to evaluate microglial function, and the limited availability of postmortem human tissue. However, the few studies that have used postmortem tissue have found changes in microglial cell number and gene expression [58,97]. This includes an increased expression of the growth arrest-specific allele 6 (GAS6) gene in microglia from humans and mice exposed to ELA [58]. Additional studies, however, are needed to further characterize ELA-induced microglial changes in human postmortem tissue using single-cell RNA-seq, spatial transcriptomics, and high-resolution confocal microscopy. This effort should be integrated into a growing recognition that microglia play a critical role in secondary-hit conditions such as depression [98,99,100,101,102] and post-traumatic stress disorder [103,104]. We suspect that some of the conflicting and inconsistent findings that plague this line of work could be explained by considering the impact of ELA in the analysis. Despite the obvious logic, we are not aware of a single study that has considered the potential role of ELA in secondary hits such as depression, PTSD, TBI, or neurodegenerative diseases.

Imaging tools to assess microglial function could be validated first in rodents and nonhuman primates where microglial activity could be manipulated. For an interesting example of this approach in rodents, see [105]. Moreover, our work demonstrates that transient ablation of microglia leads to reduced local functional connectivity, as assessed using rsfMRI [39]. This suggests that this approach may be utilized to identify deficits in microglia-mediated synaptic pruning in children and adolescents exposed to ELA. Humanized mice can also be used to test whether human microglia transplanted into mice [106] display similar abnormalities to those observed in mice exposed ELA. Alternatively, one can test whether fecal matter from children exposed to ELA [107] transplanted into germ-free mice impairs microglial-mediated synaptic pruning [49,108]. Finally, to the best of our knowledge, no studies have examined the impact of ELA on microglial function using nonhuman primate models. Such an approach will help determine whether outcomes observed in rodents, especially during postnatal development, extend to nonhuman primates.

In conclusion, studies in rodents have now demonstrated that ELA perturbs microglial function during the postnatal period. These changes affect connectivity, cognition, and response to secondary hits later in life. Few studies have also documented microglial changes in postmortem tissue in humans exposed to ELA supporting the notion that long-term alterations in these cells. Such changes may contribute to the more severe clinical presentation and the development of more specific diagnostic and effective treatments for individuals with a history of ELA.

## Figures and Tables

**Figure 1 biomolecules-14-00964-f001:**
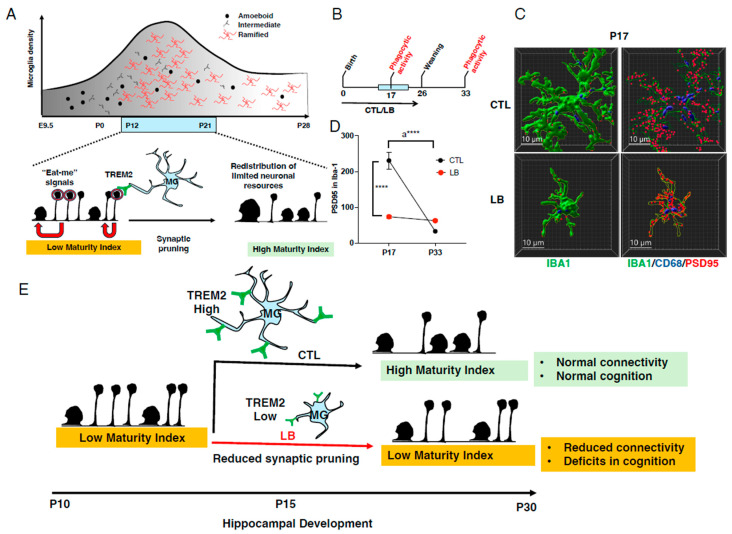
LB Impairs Synaptic Pruning During a Critical Period of Hippocampal Development. (**A**) The number of microglia and their size increase from P12 to P21 in the hippocampus (top). During this critical period, the expression of TREM2 on microglia increases, enabling the efficient removal of non-functional synapses labeled with “eat-me signals”, depicted as red circles. This process redirects limited neuronal resources to functional synapses (indicated by red arrows), enabling the formation of a more efficient network characterized by a high synaptic maturity index and improved hippocampal-dependent memory. (**B**) Mice were exposed to control (CTL) or LB conditions and perfused at P17 and P33 to assess microglial morphology and phagocytic activity. (**C**) Imaris models of P17 microglial morphology (Iba1 staining in green, left column) and synaptic engulfment (PSD95 in red, right column) and phagosome size (CD68, blue, right column). (**D**) LB reduced the number of PSD95 puncta inside microglia at P17, but not at P33. Note that phagocytic activity is eightfold higher at P17 compared to P33 in CTL, but not LB [39]. (**E**) Working model: LB impairs microglial-mediated synaptic pruning in both males and females, leading to reduced synaptic maturity, lower functional connectivity, and impaired contextual fear conditioning in adolescent males. Upregulation of synaptic pruning in astrocytes may protect female LB mice (in (**E**) but see [39] for details). **** *p* < 0.0001.

**Figure 2 biomolecules-14-00964-f002:**
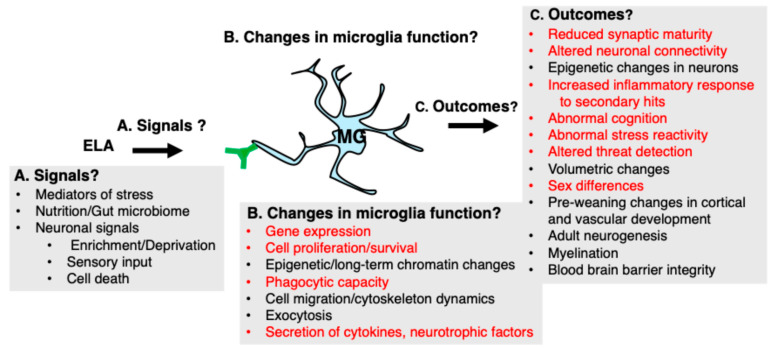
Recent Progress (in red font) and Unresolved Questions (black font). (**A**) Potential signals that drive ELA-induced microglial changes. (**B**) Alterations in microglial function. (**C**) Functional and structural changes observed in rodents exposed to ELA.

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
