# Peer review of "Microglia: The Drunken Gardeners of Early Adversity"

_biomolecules, 2024, doi:10.3390/biom14080964_

Round 1

Reviewer 1 Report

Comments and Suggestions for Authors

Ahmed et al. has done an impressive job to put things together on role of microglia in ELS associated changes in important limbic regions such as hippocampus. Although several important and comprehensive articles had published previously on interplay between microglia and stress, the current review article brings focused and deep discussion on effect of early life experiences on microglia function and the subsequent changes in neural plasticity. Here I have only a few minor comments that I believe will help to improve the quality of this manuscript.

1). In introduction or discussion, the authors can discuss in few sentences the complexity and controversial postmortem findings on role of microglia in depression. Some interesting and important review articles and research articles have been published recently indicating sex specific and non-inflammatory phenotype microglia in limbic regions such as PFC in suicide subjects. These findings also highly indicate the phagocytotic activity of microglia is affected due to depression pathology.

2). I recommend changing the title (optional though). The title is very interesting and cool for the microglia field but it might lose a more general audience when it get published.

3) I suggest merging the conclusion in future direction section.

4) it would be very interesting if the authors could discuss some similarities and differences in microglia morphology-transcriptomic between early life adversity models vs stress models in adult ( for example social defeat or chronic unpredictable stress model)

5) In figure one the section C, it is difficult to see the CD68-PSD95 colocalization. Would it possible to provide a higher resolution or magnification?

6) Would it possible to make more comments about the specificity of TREM mediated pruning during development. I was wondering if there is any report indicating complement dependent pathway or independent pathway such as CX3CR1 is impaired in animal model of ELS. At least in human it seems CX3CR1 pathway is impaired due to child abuse.

7) as the authors have had the significant contribution on effect of ELS on on hippocampal microglia, it would be great if they can provide more insight why hippocampal microglia are very sensitive and responsive to early life experiences in comparison to other brain regions. Is it a matter of regional heterogeneity?

Author Response

Please see attached document with responses to both reviewers. Thanks.

Reviewer 2 Report

Comments and Suggestions for Authors

Ahmed and colleagues present a well-written and timely review on microglia and early life adversity. Overall the review is organized well and informative. My suggestions to improve the manuscript are:

- Rodent studies using ELA models are plentiful, and certainly nuanced (e.g., variations in timing of procedures produces vastly different outcomes). There is one mention of timing for microglia migration/ pruning (E9.5 through PD14). This is a really long period developmentally, encompassing 2/3 of the preweaning age (when ELA is administered) and during which different brain regions undergo maturation at different rates (e.g., pruning vs. growth, cortical vs. subcortical structures). It would be very useful to explain in more detail, if known, when specific systems or regions are affected by microglia migration. One may expect correlations between behavioral outcomes and time-sensitive disruption of development - a more thorough discussion of this would be useful. (for example, ELA can increase susceptibility to addiction; is it the case that systems underlying addiction are maturing/ being pruned in the same time frame as microglia migration and/or ELA procedures?)

- related to data in the supplemental figure, the manuscript states that female LB littermates showed minor impairment, unlike males. However, the data look bimodal. One interpretation of that could be that some animals are resilient. It could be useful to discuss resiliency in the main manuscript. As is, the figure chosen doesn't quite seem to support the statement made about it.

- it's stated that it's 'not surprising' that LB and maternal separation produce differing effects on microglia. I find this surprising, given LB and MS are both used as ELA models. Please clarify.

Author Response

(The authors gave the same response as above.)

Round 2

Reviewer 2 Report

Comments and Suggestions for Authors

The authors have made changes to the manuscript that clarify points of concern, but an important detail was still not addressed.

Comment 2.2: The original comment was intended to convey the fact that the manuscript discusses microglia migration/pruning as taking place between E9.5 and PD14. ELA as administered in these experiments takes place only during the latter half of that time. It would be important then, to distinguish which specific microglia developmental changes are taking place only during the ELA period, which would help a reader understand what specific events may be altered by ELA. (i.e., ELA will NOT affect events taking place before birth, so which are which?) This still needs to be clarified.
